# Effect of Cadmium on Oxidative Stress Indices and Vitamin D Concentrations in Children

**DOI:** 10.3390/jcm12041572

**Published:** 2023-02-16

**Authors:** Artur Chwalba, Joanna Orłowska, Michał Słota, Marta Jeziorska, Kinga Filipecka, Francesco Bellanti, Michał Dobrakowski, Aleksandra Kasperczyk, Jolanta Zalejska-Fiolka, Sławomir Kasperczyk

**Affiliations:** 1Department of Pharmacology, Faculty of Medical Sciences in Zabrze, Medical University of Silesia in Katowice, Jordana 19, 41-808 Zabrze, Poland; 2Department of Biochemistry, Faculty of Medical Sciences in Zabrze, Medical University of Silesia in Katowice, Jordana 19, 41-808 Zabrze, Poland; 3ARKOP Sp. z o.o., Kolejowa 34a, 32-332 Bukowno, Poland; 4Centrum Medyczne MED-KOZ & MEDIKO Dąbrowski ul., Lipowa 2, 43-340 Kozy, Poland; 5Department of Medical and Surgical Sciences, University of Foggia, Viale Pinto 1, 71122 Foggia, Italy

**Keywords:** cadmium, oxidative stress, vitamin D, children

## Abstract

Heavy metal poisoning can have serious health consequences, including damage to the brain, kidneys, and other organs. Cadmium is a toxic heavy metal that can accumulate in the body over time and the exposure to this element has been linked to a variety of adverse health effects. Cadmium toxicity can lead to an imbalance in the cellular redox state and be a source of oxidative stress. On the molecular level, cadmium ions negatively affect cellular metabolism, including the disruption of energy production, protein synthesis, and DNA damage. The study has been carried out on a group of 140 school-age children (8 to 14 years old) inhabiting the industrialized areas of Upper Silesia. The study population was divided into two sub-groups based on the median concentration of cadmium in blood (0.27 µg/L): Low-CdB and High-CdB. Measured traits comprised blood cadmium levels (CdB) as well as a blood count and selected oxidative stress markers. This research study aimed to demonstrate a correlation between the impact of exposure to elevated cadmium concentrations in a population of children and certain markers of oxidative stress, and 25-OH vitamin D3 concentration. A negative correlation has been found between cadmium concentration and 25-OH vitamin D3 level, protein sulfhydryl groups content in blood serum, glutathione reductase activity, and lipofuscin and malondialdehyde levels in erythrocytes. The concentration of 25-OH vitamin D_3_ in the High-CdB group was decreased by 23%. The oxidative stress indices can be considered a valuable indicator of early Cd-toxicity effects to be included in the routinely-applied cadmium exposure monitoring parameters, allowing the evaluation of stress intensity to the cell metabolism.

## 1. Introduction

Environmental exposure to heavy metals, and cadmium in particular, constitutes a significant health problem, especially in the population of infants. However, although the incidence of acute poisoning cases has reduced, the risk of micro-intoxication persists. Therefore, it is imperative to undertake measures to either eliminate or mitigate the risk factors associated with heavy metal toxicity in children.

Air and water are capable of quick self-cleaning. Soil, on the other hand, due to its absorption properties, accumulates heavy metals over decades and even centuries. This situation is typical of certain environmental exposures including lead and cadmium, which once introduced into the environment, remain there for a long time. According to the data published by the US Agency for Toxic Substances and Disease Registry (ATSDR), approximately 25–30,000 tons of cadmium is released into the environment annually [1]. According to health risk assessments carried out in Poland, the Upper Silesia voivodeship belongs to the regions of concern in terms of Cd emission intensity (16.9% of the total emission in Poland).

Cadmium has been widely applied in different industries. It is used in the production of dyes and stabilizers for plastics, galvanic protective coatings, alkaline regular and rechargeable batteries, and fluorescent paints. Cadmium-containing compounds are also used in car tires and in agriculture as an additive to artificial fertilizers. Due to its widespread usage, cadmium has emerged as a major contributor to environmental pollution, as it does not undergo degradation once introduced into the ecosystem and exhibits a long half-life, resulting in its persistent presence in the environment [2,3]. Long-term use of cadmium leads to air, water, and soil pollution and its accumulation in plants and animal tissues [4]. Despite the significant progress in introducing stringent environmental standards and considerable advancement in environmental protections, lead and cadmium contamination constitutes a serious medical and social issue, particularly in highly industrialized regions and in developing countries [5,6].

The assimilation of cadmium compounds occurs through the digestive system or due to exposure to airborne dust. Similarly to other heavy metals, Cd ions affect the human organism at the molecular level, causing damage to tissues and organs. Cadmium accumulates in the bones, testicles, liver, and kidneys. Initially, Cd ions are absorbed by enterocytes of intestinal villi, from where they are transferred to the bloodstream and bind with erythrocytes. Subsequently, these toxic ions are transported to the liver, where they form complex bonds with cystine-rich proteins, metallothioneins (MT). The CdMT complexes may then be released into the bloodstream and further transported to the kidneys, which are the target organ of the toxic effect of cadmium in the human body. The CdMT complexes are degraded in the kidneys and Cd ions are released. This in turn leads to damaging the structures of this organ and nephropathy, resorption disorders, and eventually chronic renal failure [7]. Environmental or occupational exposure to cadmium in humans can impose chronic health effects including hematological, renal, hepatic, cardiovascular, pulmonary, musculoskeletal, neurological, and reproductive disorders. Cadmium is also classified as a probable human carcinogen, being considered as a possible causative agent of tumorigenesis in different organs, particularly the lungs. 

Cadmium compounds also interact with ions of many physiologically-relevant elements, such as zinc, copper, iron, magnesium, calcium, or selenium. As a result, the homeostasis of numerous cellular processes is disturbed. Cadmium interferes with the energy metabolism of the cell by inhibiting the biosynthesis of dehydrogenases, damaging mitochondria, and having a tendency to bond with enzymatic proteins (Figure 1). Cd ions induce oxidative stress and affect the expression of specific transcription factors [8,9]. 

### Pathomechanism of Cadmium Toxicity 

Disturbance of the flow of electrons in the respiratory chain, the release of transition metals, and impairment of antioxidant mechanisms by cadmium indirectly lead to the formation of free oxygen radicals and their reactive forms (RFT). This subsequently leads to peroxidative damage to cell membranes. Furthermore, excess RFT fosters the activation of proto-oncogenes, and cadmium ions inhibit DNA repair. On the other hand, through changes in the formation and functioning of biogenic amines and neurotransmitter amino acids, cadmium has a neurotoxic effect on the central nervous system. Therefore, the toxic effects of this element cover numerous disorders in cell metabolism, which in turn lead to morphological and functional changes in many organs [11]. Furthermore, cadmium induces and intensifies oxidative stress and increases lipid peroxidation. Cadmium may also affect the process of haematopoiesis, increasing the risk of anaemia [12,13].

A growing body of evidence demonstrates the neurotoxic effect of cadmium [14]. Specific studies have clearly shown that cadmium causes the displacement of iron and copper ions from protein molecules [15]. Cd also disturbs the flow of electrons in the respiratory chain and causes the release of transition metals [16]. 

Furthermore, cadmium causes disorders in the antioxidant system by reducing the level of reduced glutathione (GSH) in cells and decreasing the thiol status. It also affects the activity of antioxidant enzymes, such as catalase and superoxide dismutase [17,18]. Experimental studies have demonstrated that whereas short-term exposure to cadmium causes an increase in the activity of superoxide dismutase, catalase, reductase, and glutathione peroxidase, long-term exposure causes them to decrease considerably. This results from the fact that cadmium displaces the Mn, Cu, Zn, Fe, and Se ions from the active sites of antioxidant enzymes, which results in a decrease in their activity or loss of function [19].

## 2. Material and Methods

The material used in the study originated from epidemiological studies coordinated by the Medical University of Silesia in Katowice. The screening program was completed in the period 2017–2018. The study included a group of 140 school-age children from the industrialized areas of Upper Silesia who matched the following eligibility criteria: age of between 8 and 14 years, and existing records of determined blood levels of cadmium. 

The study population was divided into two sub-groups based on the median of cadmium concentration in blood (0.27 µg/L): Low-CdB (CdB level below median, mean 0.19 ± 0.05 µg/L, *n* = 69) and High-CdB (CdB value above median, mean 0.40 ± 0.11 µg/L, *n* = 71). 

### 2.1. Laboratory Procedures

Blood samples from each subject were collected from the cubital vein using tubes coated with K_3_EDTA to obtain whole blood, serum and erythrocytes.

### 2.2. Analysis of Metals, Blood Count and Vitamin D_3_ Concentration

The analysis of the level of cadmium in blood was conducted by graphite furnace atomic absorption spectrometry technique using the iCE 3400 AA Spectrometer (Thermo Fisher Scientific, Waltham, MA, USA). 

The assessment of blood morphology parameters was conducted using a Sysmex K-4500 Automated Hematology Analyzer (Sysmex Corporation, Kobe, Japan). The following measurements were obtained: white blood cells (WBC) count, red blood cells (RBC) count, hemoglobin (HGB) level, hematocrit (HCT), and platelet (PLT) count. 

Serum vitamin D_3_ concentration was determined using the Euroimmun kit 25-OH Vitamin D, ELISA nr EQ 6411-9601. The 25-OH vitamin D compounds contained in the tested sample are bound by anti-25-OH antibodies on the microplate. Free antibody binding sites are occupied by labelled 25-OH vitamin D molecules. The intensity of the colour formed after addition of the chromogen/substrate solution is measured photometrically, as colour intensity is inversely proportional to the 25-OH vitamin D concentration in the serum or plasma.

### 2.3. Markers of Oxidative Stress

The parameter of total antioxidant capacity (TAC) was measured in blood serum according to the adopted protocol of Erel [20]. In this protocol, a colored 2,2′-azinobis-(3-ethylbenzothiazoline-6-sulfonic acid) radical cation (ABTS*+) solution is decolorized by the antioxidants present in the analyzed sample. The reaction efficiency depends on a specific concentration of antioxidant compounds. The color change was measured as a change in absorbance at 660 nm using an automated analyzer. 

Total oxidant status (TOS) was measured in blood serum according to compatible methodology [21]. The measurement methodology is based on the oxidation of ferrous ions to ferric ions in the presence of various oxidant species in an acidic medium. The measurement of the ferric ion by xylenol orange is carried out using an automated analyzer.

The methodology of Arab and Steghens [22] was adopted to measure the concentrations of lipid hydroperoxides (LPH) in blood serum. In this assay oxidation of Fe II to Fe III by lipid hydroperoxides, under acidic conditions, the complexation of Fe III by xylenol orange is quantitatively detected by using an automated analyzer.

The concentration of malondialdehyde (MDA), a lipid peroxidation marker, was measured fluorometrically, in serum and erythrocytes, as a 2-thiobarbituric acid-reactive substances (TBARS) in serum according to the standard methodology of Ohkawa, Ohishi and Yagi [23]. Analyzed samples were mixed with 8.1% sodium dodecyl sulfate, 20% acetic acid, and 0.8% 2-thiobarbituric acid. After vortexing, samples were subsequently incubated for 1 h at 95 °C and butanol-pyridine 15:1 (*v/v*) solution was added. The mixture was shaken for 10 min and then centrifuged. The butanol-pyridine layer was tested fluorometrically at 515 nm and 522 nm excitation wavelengths (PerkinElmer, Waltham, MA, USA).

The concentrations of lipofuscin (LPS) were tested according to the methodology of Tsuchida et al. [24]. Measurements were expressed in serum in the relative unit [RU/L] and in erythrocytes [RU/g] Hb.

The concentrations of protein sulfhydryl groups (PSH) were evaluated in accordance with the method developed by Koster, Biemond, and Swaak [25]. The reaction rate of DTNB, undergoing reduction of protein-bound compounds containing thiol groups, was tested by the yield of the yellow anion derivative, 5-thio-2-nitrobenzoate, which absorbs at a wavelength of 412 nm with the use of a PerkinElmer automated analyzer (PerkinElmer, Waltham, MA, USA).

The level of ceruloplasmin (CER) concentration in blood serum was analysed in accordance with the protocol described by Richterich [26]. Measurements were conducted using a PerkinElmer automated analyzer (PerkinElmer, Waltham, MA, USA).

### 2.4. Antioxidant Enzymes

For the assessment of the activity of superoxide dismutase (SOD) in blood serum and erythrocytes, the methodology of Oyanagui [27] was applied. Activities of the superoxide SOD enzyme were expressed in nitric units (NU) in each mL as [NU/mL] (serum) and [NU/mg] Hb (erythrocytes). The activity of catalase (CAT) in erythrocytes was evaluated in accordance with the method of Johansson and Håkan Borg [28]. The activity of glutathione reductase (GR) in erythrocytes was tested according to Richterich [26], glutathione S-transferase (GST) according to Habig and Jakoby [29], and glutathione peroxidase (GPx) according to Paglia and Valentine [30]. Measurements were conducted using a PerkinElmer automated analyzer (PerkinElmer, Waltham, MA, USA). 

### 2.5. Statistical Analysis

Statistical analysis was performed using Statistica 12.0 PL software (StatSoft Polska, Kraków, Poland). Descriptive statistics were reported as mean ± standard deviation (SD) for normal distribution. An initial Shapiro-Wilk test was used to determine the normality of distribution of each tested variable, and Levene’s test was applied to verify the homogeneity of variances. Statistical comparisons were carried out with an application of Student’s *t*-test, Mann-Whitney U test, and chi-squared test. Spearman’s rank-order correlation coefficients were calculated for the assessment of existing associations between tested variables. PCA (Principal Component Analysis) was performed for the recognition of existing associations of more than two components. Additionally, regression analysis was performed (R—multiple correlation coefficient, R2 coefficient of determination, β*-regression standardized coefficient). A probability at *p* ≤ 0.05 was considered statistically significant.

The research project was approved by the Bioethics Committee of the Medical University of Silesia in Katowice (KNW/0022/KB1/108/14). In funding terms, this work was supported by the Medical University of Silesia in Poland and zinc smelter Miasteczko Śląskie.

## 3. Results

In the subpopulation of children with higher cadmium levels, higher levels of TAC (total antioxidant capacity) were observed, along with decreased content of sulfhydryl groups (SH) and a divergent effect on the oxidative stress parameters. The level of malondialdehyde (MDA) in erythrocytes was significantly lower and a concentration of lipofuscin (LPS) in serum was higher. An increased activity of glutathione peroxidase (GPX) when compared to children with lower cadmium levels was also observed (Table 1 and Table 2). Blood count evaluation had no significant differences between the examined groups (Table 3).

The analysis of cadmium concentrations in the evaluated subgroups revealed a statistically significant positive correlation with the levels of TAC, LPS in blood serum, and GPX activity in erythrocytes. The negative correlations of cadmium levels were confirmed for vitamin D_3_, SH group content in blood serum, and GR activity, and LPS and MDA levels in erythrocytes were also demonstrated (Table 4). 

PCA (Principal Component Analysis) computations were carried out in order to better understand the joint effects of cadmium exposure markers, statutory vitamin D3 concentration and antioxidative status indices. Preliminary analyses resulted in the selection of the following principal components (PCs): PC1—blood cadmium concentrations [µg/dL]; PC2—concentration of lipofuscin (LPS); PC3—25-OH vitamin D concentration [ng/mL]. The 3D-PCA analysis conducted proved the distinctiveness of pre-defined subgroups that were classified based on the Cd level above or below the median value. Analysis of the tested variables over the 3D matrix demonstrated a clear differentiation of the Low-CdB subgroup (red dots on the plot), that was characterized by a lower Cd level (as per the study design), lower vitamin D concentration and lower LPS concentration, whereas the High-CdB group (green dots on the plot) was defined by higher Cd levels, lower vitamin D concentration and higher LPS value (Figure 2).

Regression analysis showed that among the parameters which are associated with changes in cadmium concentration were vitamin D concentration (β* = −0.34), content of sulfhydryl groups (SH) (β* = −0.19), and a concentration of lipofuscin (LPS) in serum (β* = 0.16) (R = 0.46, R^2^ = 0.21, *p* < 0.001 for above model).

## 4. Discussion

Heavy metals affect the oxidative stress status of an organism as they contribute to the generation of oxygen free radicals and cause modifications to the functioning of the antioxidant system. It was demonstrated that Cd ions cause the displacement of iron and copper ions from the specific protein molecules [15]. Cadmium can also cause disturbances in the antioxidant system by reducing the level of reduced glutathione (GSH) in cells and decreasing the thiol status. Cd ions also directly affect the activity of antioxidant enzymes, such as catalase and superoxide dismutase [17,31]. It has been demonstrated, however, that while short-term exposure to cadmium causes an increase in the activity of superoxide dismutase (SOD), catalase (CAT), and glutathione peroxidase (GPX), long-term exposure can cause the constitutive decrease. This can be attributed to the fact that cadmium displaces the physiological metals in the active sites of antioxidant enzymes, causing their malfunction [14,19].

A recent study was focused on a comprehensive assessment of the impact of prenatal and early childhood exposure to cadmium [32]. This aspect was also investigated in other trials [33]. Exposure to cadmium in the prenatal period may have a negative effect on the fetus, as according to the studies, cadmium penetrates through the placental barrier and its level in the umbilical cord blood may constitute 10% to 70% of its level in the mother’s blood [34]. There have been several studies focusing on the toxic effects of cadmium on unborn children. Menai et al. demonstrated its influence on fetal growth inhibition and the birth weight of the child [18]. These findings were also confirmed by the research demonstrating the presence of higher cadmium levels in the placentas of newborn babies with lower birth weight as compared to the control group with normal body weight [35]. It has been demonstrated that a prenatal exposure to cadmium may cause growth retardation in early childhood [36].

Heavy metals enter the body through the digestive or respiratory systems. Specific childhood behavioral patterns constitute an important exposure factor. Nevertheless, the main source of exposure is associated with food intake, particularly vegetables and fruit originating from the contaminated areas. Children absorb more of this metal compared to adults. In children, absorption through the lungs is several times higher than through the digestive system. This can be associated with the greater lung capacity and physical activity of children, which favour greater absorption of environmental toxicants [37].

Another important determinant of higher absorption of heavy metals is sex. Among the examined children, girls demonstrated higher levels of accumulated cadmium. This observation has also been confirmed in earlier studies [38].

Cadmium demonstrates a negative effect on renal function, interfering with the liver metabolism, as well as causing genotoxicity effects. The effects of Cd exposure depend on the dose, pathway, and duration of the exposure [9]. The authors presented a review of signal pathways and mechanisms involved in the modulation of cadmium toxicity. Extensive descriptions of the toxic effect of cadmium on individual organs and systems in the human body have also been provided in earlier scoping reviews [39].

Symptoms of acute cadmium poisoning include fever, shortness of breath, and fatigue, as well as pulmonary edema and pneumonia. In severe cases of Cd poisoning, this may lead to respiratory failure and death.

Multiple studies have established cadmium as a carcinogen. A thorough examination of the relevant literature has recently been presented [2,6,40,41]. According to the International Agency for Research on Cancer (IARC), cadmium is classified as belonging to group 1, covering substances that are carcinogenic for humans. Recently-published results from a study conducted in the Chinese population investigated the impact of prolonged cadmium exposure on blood glucose concentration levels and the associated risk of diabetes [42]. Based on these findings, the authors concluded that long-term exposure to cadmium for adults in the general population may contribute to increased levels in glucose levels in the blood and lead to the development of type 2 diabetes.

It was demonstrated that Cd can interfere with vitamin D metabolism. Vitamin D increases intestinal calcium and phosphate absorption and stimulates the co-absorption of other essential minerals, such as magnesium, iron, and zinc. It also influences the uptake rates of other toxic metals including lead, cadmium, aluminium, and cobalt as well as radioactive isotopes such as strontium and caesium. Vitamin D concentration can also be considered an important factor affecting the toxicity of these toxic metals by increasing their absorption and retention. Cadmium interferes with normative vitamin D metabolism by blocking the renal synthesis of 1,25-dihydroxy vitamin D [43,44]. The presented study provided additional evidence while demonstrating a reduced concentration of 25-OH vitamin D_3_ in the High-CdB group. Some pharmacological studies suggest that supplementation with vitamin D can minimize the toxic Cd effect in populations exposed to Cd [45]. A chronic cadmium exposure has also been associated with osteotoxicity in adults [44]. Moreover, researchers have demonstrated the alleviated risk of osteoporosis in individuals exposed to cadmium [46].

The presented study aimed to demonstrate a direct relationship between the effects of exposure to higher cadmium levels in children and certain markers of oxidative stress. The research proved the higher TAC levels and lower sulfhydryl groups (SH) content in the tested population. The Cd-toxicity effect on oxidative stress indices was also demonstrated by a lower content of malondialdehyde in erythrocytes, a higher content of lipofuscin in serum, and higher activity of glutathione peroxidase when compared to children with lower cadmium concentrations. These differences in serum and blood cell concentrations may be associated with the damage to cell membranes caused by Cd ions [6,7].

## 5. Conclusions

The presented study demonstrated the significant relationship between the effect of exposure to higher cadmium levels in children and oxidative stress indices. The main advantage of the study is the fact that it investigates a group of children residing in a clearly-defined region, characterized by environmental exposure to cadmium and lead compounds. This facilitated a comprehensive analysis of the socioeconomic and environmental factors affecting the concentrations of these metals in children’s blood, and their impact on blood count parameters, oxidative stress, and vitamin D_3_ levels. Unfortunately, the study had some limitations. Due to its retrospective character, it was impossible to clarify some survey questions in more detail or fully verify the answers given, hence not allowing for in-depth analysis. Likewise, it was impossible to take into account the potential intake of dietary supplements or vitamin preparations. Other factors that might influence levels of heavy metals, such as air, soil, and potable water contamination, were not studied either.

The results of the presented study highlight the potential risk sources for the child population of Upper Silesia and emphasize the necessity of continuing measures aimed at limiting risk factors for heavy metal poisoning, supporting social campaigns, and further education of the most vulnerable populations.

## Figures and Tables

**Figure 1 jcm-12-01572-f001:**
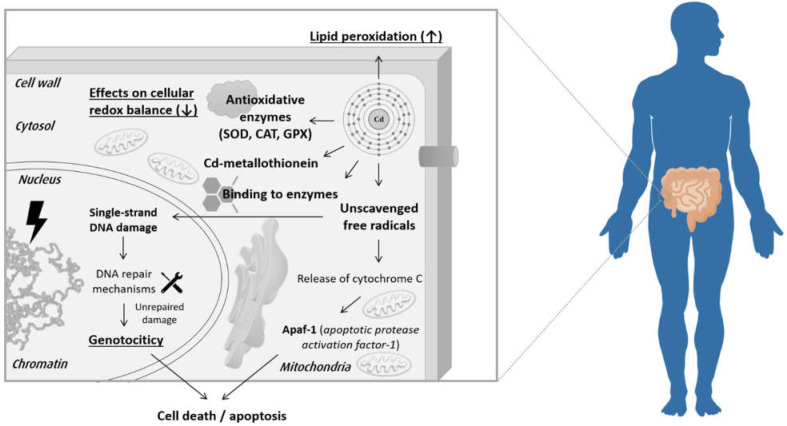
Overview of molecular mechanisms of cadmium toxicity and detoxication pathways in humans (based on [10]).

**Figure 2 jcm-12-01572-f002:**
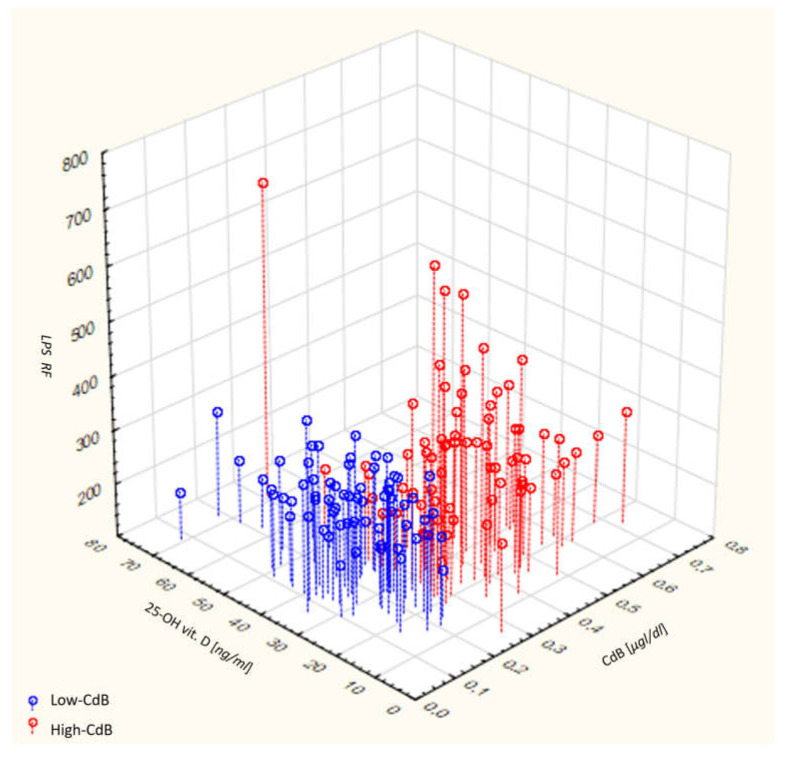
A three-dimensional PCA plot generated from principal component analysis (PCA) of the quantitative variables related to blood cadmium concentrations [µg/dL], concentration of lipofuscin (LPS), and 25-OH vitamin D concentration [ng/mL]. The blue dots indicate the data points corresponding to the Low-CdB (Cd level below median) subgroup, whereas the red dots correspond to the High-CdB (Cd value above median) subgroup.

**Table 1 jcm-12-01572-t001:** Characteristics of the studied population, concentrations of the analyzed metals and vitamin D_3_ divided into groups depending on the blood cadmium levels. Low CdB (CdB level below median) and High CdB (CdB value above median). Statistically significant values are marked in bold type. BMI—Body Mass Index.

	Low CbB Group		High CdB Group (above 0.27 µg/L)		Change %	*p* Value
	(below 0.27 µg/L)					
	*n* = 69		*n* = 71			
	Mean	SD	Mean	SD		
Body mass	33.9	7.52	37.1	10.37	9%	0.069
Height	137.1	7.49	141.1	8.91	3%	0.004
BMI	17.9	2.92	18.3	3.57	2%	0.425
Age	9.8	1.00	10.28	1.04	5%	0.001
Sex (percentage of boys)	54%		37%			0.044
Cadmium concentration—CdB (µg/L)	0.19	0.05	0.40	0.11	115%	<0.001
Magnesium concentration (mg/dL)	2.18	0.14	2.20	0.13	1%	0.599
Calcium concentration (mg/dL)	102	4.65	104	4.53	1%	0.074
Iron concentration (µg/dL)	87.1	37.4	87.7	36.8	1%	0.868
25-OH Vitamin D concentration (ng/mL)	35.8	12.7	27.6	10.2	−23%	<0.001

**Table 2 jcm-12-01572-t002:** Oxidative stress indexes in serum and erythrocytes divided into groups depending on blood cadmium concentrations. Low CdB (CdB level below median) and High CdB (CdB value above median). Statistically significant values are marked in bold type. PSH—protein sulfhydryl groups; CER—ceruloplasmin; TAC—total antioxidant capacity; TOS—total oxidant status; LPH—lipid hydroperoxides, SOD—superoxide dismutase; LPS—lipofuscin; MDA—malondialdehyde; GR—glutathione reductase; CAT—catalase; SOD—superoxide dismutase; GPX—glutathione peroxidase; GST—glutathione S-transferase.

	Low CdB Group		High CdB Group		Change %	*p* Value
	(below 0.27 µg/L)		(above 0.27 µg/L)			
	*n* = 69		*n* = 71			
	Mean	SD	Mean	SD		
PSH group content µmol/g of protein	5.08	0.66	4.77	0.49	−6%	0.004
CER concentration mg/dL	33.6	7.22	35.6	7.68	6%	0.246
TAC mmol/L	0.97	0.11	1.03	0.09	6%	<0.001
TOS umol/L	4.02	2.54	3.74	2.58	−7%	0.406
LPH—lipid hydroperoxides umol/L	1.83	1.35	1.72	1.07	−6%	0.809
SOD activity NU/mL	20.6	2.71	20.6	2.77	0%	0.746
LPS concentration RF	264	46.1	312	108.9	18%	0.001
MDA concentration µmol/L	1.79	0.44	1.7	0.46	-5%	0.2
GR activity IU/g Hb	8.58	2.25	7.91	2.55	-8%	0.116
CAT activity IU/g Hb	531	90.4	510	108	−4%	0.102
SOD activity NU/mgHb	227	27.3	222	25.6	−2%	0.144
GPX activity IU/gHb	55.5	18.1	75.78	20.4	37%	<0.001
GST activity IU/gHb	0.28	0.14	0.27	0.16	−4%	0.465
LPS concentration RF/gHb	1235	354	1138	360	-8%	0.111
MDA concentration µmol/gHb	0.53	0.07	0.47	0.07	−10%	<0.001

**Table 3 jcm-12-01572-t003:** Blood count parameters divided into subgroups depending on cadmium concentrations in the blood: Low CdB (CdB level below median) and High CdB (CdB value above median). WBC—white blood cells; RBC—red blood cells; HGB—hemoglobin; HCT—hematocrit; PLT—platelets.

	Low CdB Group		High CdB Group		Change %	*p* Value
	(below 0.27 µg/L)		(above 0.27 µg/L)			
	*n* = 69		*n* = 71			
	Mean/%	SD	Mean/%	SD		
WBC	6.84	1.4	6.54	1.53	−4%	0.229
RBC	4.86	0.31	4.86	0.3	0%	0.947
HCT	40	2.25	40	1.9	0%	0.926
HGB	13.6	0.8	13.7	0.62	1%	0.56
PLT	318	53.1	312	66.7	−2%	0.552

**Table 4 jcm-12-01572-t004:** Analysis of the correlation between cadmium (CdB) and the investigated parameters (Spearman R correlation coefficient). PSH—protein sulfhydryl groups; CER—ceruloplasmin; TAC—total antioxidant capacity; TOS—total oxidant status; LPH—lipid hydroperoxides; SOD—superoxide dismutase; LPS—lipofuscin; MDA—malondialdehyde; GR—glutathione reductase; CAT—catalase; SOD—superoxide dismutase; GPX—glutathione peroxidase; GST—glutathione S-transferase.

	R	*p*
Magnesium concentration (mg/dL)	0.1	0.262
Calcium concentration (mg/L)	0.19	0.028
Iron concentration (ug/dL)	0.03	0.719
Vitamin D_3_ concentration (ng/mL)	−0.37	<0.001
PSH group content µmol/g of protein	−0.3	<0.001
CER concentration mg/dL	0.09	0.306
TAC concentration mmol/L	0.33	<0.001
TOS concentration umol/L	−0.1	0.241
Peroxide level (LPH) umol/L	−0.06	0.499
SOD activity NU/mL	0.08	0.337
MnSOD activity NU/mL	−0.06	0.455
CuZnSOD activity NU/mL	0.11	0.182
LPS concentration RF	0.25	0.003
MDA concentration µmol/L	−0.16	0.066
GR activity IU/g Hb	−0.18	0.031
CAT activity IU/g Hb	−0.11	0.184
SOD activity NU/mg Hb	−0.1	0.222
GPX activity IU/g Hb	0.55	<0.001
GST activity IU/g Hb	−0.11	0.207
LPS concentration RF/g Hb	−0.23	0.007
MDA concentration µmol/g Hb	−0.45	<0.001

## Data Availability

The data presented in this study are available on request from the corresponding author. The data are not publicly available due to privacy or ethical restrictions.

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
