# Peer review of "Effect of Cadmium on Oxidative Stress Indices and Vitamin D Concentrations in Children"

_jcm, 2023, doi:10.3390/jcm12041572_

Round 1
Reviewer 1 Report
In the current work, the authors investigated effect of cadmium on oxidative stress indices and vitamin D concentrations in children. The study is interesting but most topics have been previously discussed. The novelty seems not to be adequate in this journal (JCM). Another problem is that the English structure needs to be corrected more in the revised manuscript.
Author Response
The subjected original study aims to elucidate the aspect of relationship between the effect of exposure to higher cadmium levels in children with the oxidative stress indices and vitamin D statutory concentrations. In order to better relate to the hypothesis of the study, additional statistical analysis was performed to illustrate the complexity of factors in the clinical picture and to identify the diagnostic potential of monitoring vitamin D concentration as a marker for assessing the toxic effect of exposure to cadmium.
Answering the reviewer’s above remarks, the whole manuscript has been cross-checked by the authors in order to improve any grammatical and stylistic errors. The final version of manuscript has been submitted to a proofreading by a native speakers to maintain the quality of the English grammar.
An inter-university group of authors, with primary affiliation to the Department of Biochemistry, Medical University of Silesia (group leader: Professor Sławomir Kasperczyk) retain a leading expert position in the field of environmental and occupational heavy metal exposure. Authors have published >100 original peer-reviewed studies in the period of last 15 years thematically related to HM toxicity monitoring, exposure risk assessment and evaluation of long term effects on human health.
Reviewer 2 Report
This paper represents correlation of cadmium in children and parameters of oxidative stress. it is well written paper, but as it is said in this manuscript the major problem is that these results are not the original resuts but they have been takend from "The material used in the study was originating from epidemiological studies routinely carried out by the “Miasteczko ÅšlÄ…skie” Foundation for Children in 2017-2018." - line 125-126
So this is not original paper.
There are also some minor improvment -
Why the cadmium is express in in microg/L?
Why the cadmium was taken from blood not from urine>
There are different fonts in whole paper. There are fullstop missing.
line 166 - SOD is express in some unusual way. Why?
Table 2. - CAT not KAT.
- SH group content - in Table 2? What is the exepriment that it has been done for that?
Author Response
Author’s present an original research study coordinated by Medical University of Silesia in Katowice that was conducted during a routine screening program in the period of 2017-2018. The confusing statement from the Materials & Methods section that provided an affiliation to “Miasteczko ÅšlÄ…skie” Foundation for Children (that is a partnering NGO involved in the health monitoring programs in the Cd-exposure areas of Upper Silesia region) has been excluded.
Answering the reviewer’s above remarks, the whole manuscript has been cross-checked by the authors in order to improve any grammatical and stylistic errors. Fonts used in the manuscript has been standardized.
The minor errors in the Table. 2 has been corrected. Protein sulfhydryl groups (PSH) were evaluated in accordance with the method by Koster, Biemond, and Swaak (cited in the methodology section).
Reviewer 3 Report
The authors have developed an interesting manuscript analyzing the effect of cadmium on a human cohort of children. In order to improve the quality of the article, I propose the following suggestions:
- After the introduction, the section "Pathomechanism of cadmium toxicity" alters the typical structure of an academic article. The authors could consider including the information in the discussion itself by contextualizing the results.
- The basis of the study depends on the mean (median) concentration of cadmium in the study cohort. However, the small sample size compared to other cohorts raises doubts about the cut-off point studied. To solve this, I suggest that the observed concentrations be compared with other articles in the academic literature.
- Statistical analysis could be a bit more complex including multiple stepwise regression analysis. I suggest that the authors consider further statistical analysis.
Kind regards.
Author Response
The structure of the text has been revised in order to better match the style and concept of the article.
In order to better relate to the hypothesis of the study, additional statistical analysis (PCA and regression analysis) was performed to illustrate the complexity of factors in the clinical picture and to identify the diagnostic potential of monitoring vitamin D concentration as a marker for assessing the toxic effect of exposure to cadmium.
Reviewer 4 Report
It is a retrospective study which underlined the effects of cadmium toxicity on oxidative stress and essential elements of body in children aged 8-14 years. The present study reported that the high cadmium level in blood was correlated with low level of vitamin D3 in study participated children.
Researches have been addressing the issue of the effects of exposure to cadmium in the prenatal period, in early childhood and preschool children. Therefore, the present study has the originalty due to participant ages.
Additionally, girls demonstrated higher levels of cadmium which has been confirmed in earlier studies.
Some corrections are needed:
-Lines 183-185 should be placed after conclusion section.
-There are different writing charachters through the MS. They should be correct as Palatino Linotype
Author Response
The structure of the text has been revised in order to better match the style and concept of the article. Fonts used in the manuscript has been standardized.
Round 2
Reviewer 1 Report
In general, the revised manuscript has the evidence and information to the readers. But the picture of Figure 1 is still blurred. This should be made the correction.
Author Response
According to the reviewer's remarks, the original figure no. 1 has been replaced with a less compressed graphical file. A full-resolution .tiff and/or .pdf version of the illustration will be provided to the editorial team in the manuscript completion process. To match the exact word count criteria for this kind of article, additional elaboration on the methods used in the study has been made. The final word count in the revised version is 4021.